# Thermal detection of single photons using Dirac fermions

Bevin Huang[1,12], Ethan G. Arnault [2,12], Woochan Jung[3,12], Caleb Fried [2], B. Jordan Russell[4,5], Kenji Watanabe [6], Takashi Taniguchi [7], Erik A. Henriksen [4,5], Dirk Englund [2], Gil-Ho Lee [3] ✉ & Kin Chung Fong [8,9,10,11] ✉

Detecting single photons is a crucial process in quantum science, quantum networking, biology, and advanced imaging. To detect the small quantum of energy carried in a photon, conventional mechanisms rely on energy excitation across either a semiconductor bandgap or superconducting gap that hinders their applications to low-energy photons. Here, we detect single near-infrared photons using the thermal properties of Dirac fermions in graphene. By exploiting the extremely low heat capacity of Dirac electrons near its charge neutrality point, we observe a temperature rise up to ~ 2 K using a hybrid Josephson junction. In this proof-of-principle experiment, we achieve an intrinsic quantum efficiency of 87% (75%) with dark count < 1 per second (per week), reaching an effective noise equivalent power of $2 \times 10^{-22}$ W/$\sqrt{\text{Hz}}$. The highest operation temperature is 1.2 K. Our results highlight the potential of graphene bolometers for detecting lower-energy photons from the mid-IR to microwave regimes, opening pathways to study space science in far-infrared regime, to potential applications in dark matter searches, and to advance quantum technologies across a broader electromagnetic spectrum.

Photons are the quantum particles of electromagnetic field, each carrying a minuscule amount of energy. This makes their detection, particularly at lower energies, challenging. Most conventional single-photon detectors (SPDs)[1–5] operate by a photo-excitation across an energy gap. In semiconductor-based avalanche photodiodes[6], the excitation creates an electron-hole pair across the bandgap. In superconducting nanowires[7–9] or kinetic inductance detectors[10], the excitation breaks Cooper pairs and promotes quasiparticles above the superconducting gap, $\Delta_s$. In each case, the energy gap provides a

mechanism to distinguish photons from dark counts caused by fluctuations, but also limits the detection of lower-energy photons[10,11]. Thermal detection of single photons (e.g., transition-edge sensors[12]) can potentially resolve this dilemma. However, owing to large electron densities and electron-phonon coupling, these SPDs typically have a substantial heat capacity, limiting their efficacy in detecting lower-energy photons.

Graphene presents a promising material for single-photon bolometers (SPB)[13–17]. Specifically, the vanishing density of states of Dirac

[1]Intelligence Community Postdoctoral Research Fellowship Program, Massachusetts Institute of Technology, Cambridge, MA, USA. [2]Department of Electrical Engineering and Computer Science, Massachusetts Institute of Technology, Cambridge, MA, USA. [3]Department of Physics, Pohang University of Science and Technology, Pohang, Republic of Korea. [4]Department of Physics, Washington University in St. Louis, St. Louis, MO, USA. [5]Center for Quantum Leaps, Washington University in St. Louis, St. Louis, MO, USA. [6]Research Center for Electronic and Optical Materials, National Institute for Materials Science, Tsukuba, Japan. [7]Research Center for Materials Nanoarchitectonics, National Institute for Materials Science, Tsukuba, Japan. [8]RTX BBN Technologies, Quantum Engineering and Computing Group, Cambridge, MA, USA. [9]Present address: Quantum Materials and Sensing Institute, Northeastern University, Burlington, MA, USA. [10]Present address: Department of Electrical and Computer Engineering, Northeastern University, Boston, MA, USA. [11]Present address: Department of Physics, Northeastern University, Boston, MA, USA. [12]These authors contributed equally: Bevin Huang, Ethan G. Arnault, Woochan Jung. ✉e-mail: lghman@postech.ac.kr; k.fong@northeastern.edu

electrons near the charge neutrality point results in a very low electronic specific heat ( ~ 1 $k_B/\mu m^2$ with $k_B$ being Boltzmann constant) and suppressed electron-phonon (E-Ph) coupling[18,19]. Consequently, the energy from a single photon can enormously raise the graphene electron temperature, $T_e$, for SPB. Yet, utilizing Dirac fermions as a bolometric material has its own challenges. For instance, the fleeting $T_e$ rise requires simultaneously fast and accurate readouts to measure photon absorption. Moreover, infrared photons may interact directly with the superconducting electrodes, generating quasiparticles that interfere with the operation of the thermal sensor. Despite remarkable progress in achieving graphene bolometers with sensitivities at the fundamental thermodynamic limit[16,17], a Dirac-fermion SPB remains elusive. In this work, we implement an optical scanner at cryogenic temperatures. We demonstrate that, upon absorption, the internal energy from a single photon can heat up the electrons and propagate through the graphene, thermally triggering the switching of a Josephson junction[15,20]. The thermal properties of Dirac fermions in graphene enable SPB to simultaneously achieve high quantum efficiency and low dark counts, yielding an effective noise equivalent power (NEP) of $2 \times 10^{-22}$ W/$\sqrt{\text{Hz}}$ that is comparable to state-of-the-art near infrared SPDs (comparison table in SI).

## Results and Discussion
### Experimental Setup

Figure 1A depicts our setup in a dilution refrigerator at temperatures $T_0 \simeq 20$ mK. 1550 nm light is routed through a single-mode optical fiber into a collimator and subsequently focused by an aspherical lens. This optical set-up is affixed on top of a three-axis piezoelectric stage which can steer the highly attenuated laser source from room temperature to the graphene absorber of area $4\,\mu m \times 25\,\mu m$ (Fig. 1B) with sub-$\mu m$ spatial precision and a beam spot size of 4 $\mu m$. We can scan over the device and measure the laser reflectometry signal, $V_{\text{refl}}$, (Methods) to identify features on the chip and ensure the location of our beam spot over the graphene (Fig. 1C). Upon absorption, the single photon will create a hotspot of heated electrons[21], which will quickly diffuse across the graphene[22], dissipating energy to the graphene lattice via E-Ph coupling. When diffusion dominates over E-Ph dissipation, the entire graphene area reaches a uniform $T_e$ that peaks at $T_{1p} = \sqrt{2h\nu/\gamma_S \mathcal{A} + T_0^2}$ ref. 15 with $h$ being the Planck constant, $\nu$ photon frequency, $\mathcal{A}$ the graphene area, and $\gamma_S$ Sommerfeld constant, which is the ratio of the electronic specific heat per unit area, $c_e$, to $T_e$. For the graphene in Fig. 1B, $T_{1p} \sim 2$ K. To overcome the challenge of measuring the rise of $T_e$ in a short time scale of a few tens of ns[16], we use a graphene-based

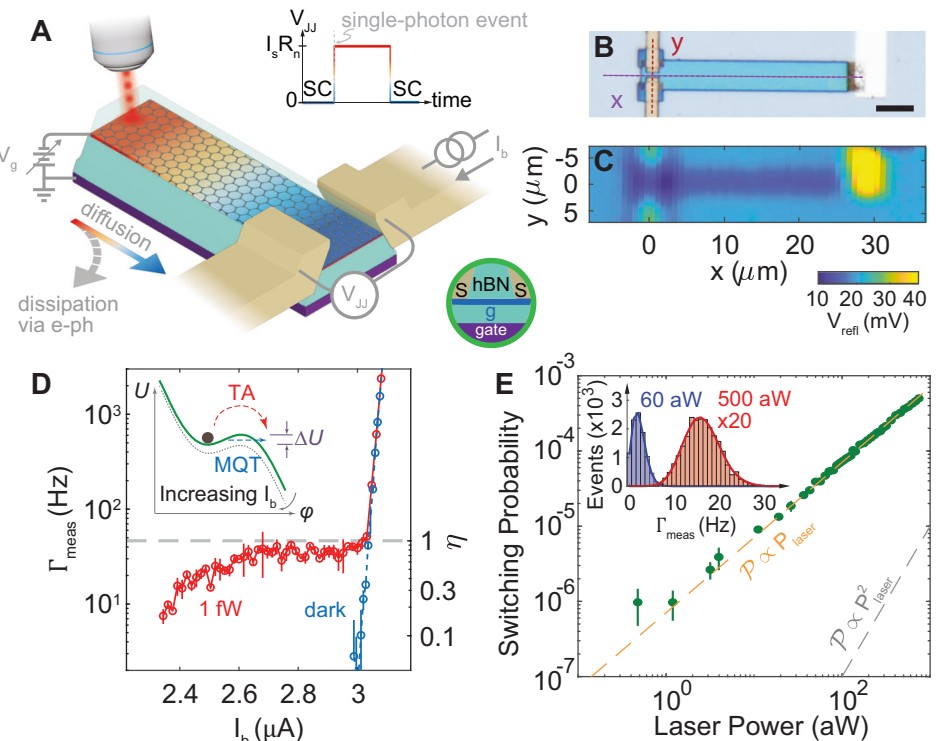

**Fig. 1 | Dirac-fermion single-photon bolometer. A** Illustration of the experiment. A photon is absorbed on one end of the graphene heating the electrons. The hot electrons then diffuse throughout the graphene while dissipating into the lattice via electron-phonon coupling. This diffusion and dissipation is determined by the geometry of the device, the base temperature and the electron density (which is controlled by an electrostatic backgate voltage, $V_g$. Inset: Junction voltage $V_{JJ}$ as a function of time. When the photon is absorbed, the junction switches from superconducting (SC) to resistive causing a voltage drop across the junction corresponding to the product of the switching current, $I_s$ and normal state resistance $R_n$. The device is then reset to the superconducting state. **B** Optical image of one of the Dirac-fermion single photon bolometers (SPBs). Scale bar is 5 $\mu m$. **C** The 2D reflectometry measurement of the device at low temperature. **D** Switching rate $\Gamma_{\text{meas}}$ vs. current bias $I_b$ for laser on (red) and off (blue). The mean value of quantum efficiency, $\eta$, in the plateau from 2.7 $\mu A$ < $I_b$ < 3 $\mu A$ with 1 fW of laser power is 0.77

± 0.08. Error bars correspond to the standard deviation in $\Gamma_{\text{meas}}$ over multiple runs. Inset: Current-biased Josephson junction (JJ) can be described as a macroscopic quantum phase particle (of phase $\varphi$) subjected to a tilted-washboard potential in the Resistively Capacitance Shunted Junction model. When dark, the junction is nominally in the macroscopic quantum tunneling (MQT) regime, however the photon raises the temperature of the junction causing a thermally activated (TA) switching event over the barrier potential $\Delta U$. **E** Junction switching probability vs. laser power at $V_g = 2$ V and $I_b/\langle I_s \rangle \simeq 0.87$. The junction switching probability is linearly proportional to the laser power (see orange dashed line highlighting the linear trend), confirming a single photon can switch the Josephson junction from superconducting to resistive. Deviation from the linear trend at lower powers is due to dark counts. Inset: Histogram of switching events with 60 aW (blue) and 500 aW (red) of laser powers adhere to Poissonian statistics. Error bars correspond to the standard deviation in switching probability over multiple runs.

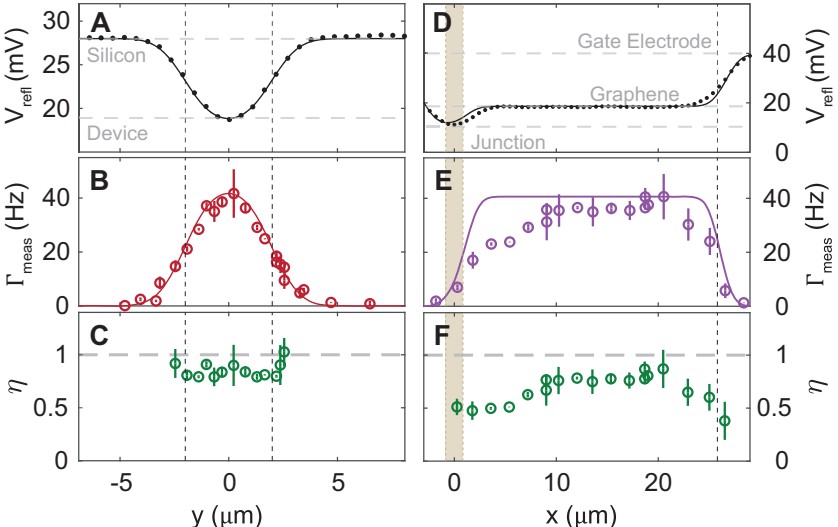

**Fig. 2 | Scanning the beam spot across the graphene. A–C** Scanning in the transverse ($y$-) direction marked by the maroon dashed line in Fig. 1B. Vertical dashed lines mark the graphene location. **D–F** Scanning in the longitudinal ($x$-) direction. Vertical light yellow box designates the Josephson-junction location. Vertical dashed line marks the graphene location. **A, D** Reflectance signal (dots) and fitting to the convolution integral (solid line). The dashed gray lines indicate the calculated reflectance values of silicon and the graphene heterostructure. **B, E** The measured and expected switching rate (open dots and solid line, respectively) based on the convolution between graphene and a Gaussian beam of spot size 4 $\mu m$. **C, F** Quantum efficiency calculated by dividing $\Gamma_{meas}$ with the absorbed photon rate. The mean value of $\eta$ in (**C**) is $0.85 \pm 0.07$. All data were taken at $I_b/\langle I_s \rangle \simeq 0.87$ and at backgate voltage, $V_{gate} = 2$ V. Error bars represent the standard deviation over multiple runs.

Josephson junction (GJJ) (fabrication details in Methods, and a table of all relevant device parameters, such as junction dimensions and flake thickness can be found in Supplementary Table S1), whose response rate is on the order of the plasma frequency[20], $\omega_p \gtrsim 100$ GHz, for the SPB readout.

We can use the Resistively and Capacitively Shunted Junction (RCSJ) model[23] to understand how a single photon switches the GJJ. In RCSJ, a macroscopic quantum phase particle with a phase difference, $\varphi$, between the two superconducting electrodes is subject to a washboard potential (Fig. 1D inset). When the phase particle is trapped initially in a local minima, i.e., $d\varphi/dt = 0$, the voltage drop across the GJJ is zero. The bias current, $I_b$, running through the GJJ tilts the washboard potential and the phase particle stochastically escapes from the minimum. When it escapes, either by thermal activation (TA)[24] over or macroscopic quantum tunneling (MQT)[25] through the barrier, $\Delta U$, the voltage drop across the GJJ becomes finite and the GJJ switches to the normal resistive state at a switching current $I_s$ (Fig. 1A inset). When the phase particle is retrapped at a retrapping current $I_r$, the GJJ switches back to the supercurrent state. The hysteretic behavior, i.e., $I_s > I_r$, frequently observed in graphene-based GJJs due to self-Joule heating[16,26,27], is useful to our investigation. When the GJJ latches into the resistive state after switching, we register a click, reset the bias current, and over time, measure the switching statistics[28] under different light intensities, densities of graphene electrons, and temperatures.

**Single-Photon Detection**

Figure 1D shows the measured switching rate, $\Gamma_{meas}$, versus $I_b$. We designate $\Gamma_{meas}$ without photons as the dark count rate, $\Gamma_{dark}$, which is governed by quantum fluctuations. The fit (dashed line) appears nearly straight in the log-linear plot because the rate of MQT follows activation theory, i.e., $\propto \exp(-7.2\Delta U/\hbar\omega_p)$[25], where $\hbar$ is the reduced Planck constant and $\hbar\omega_p$ is the zero-point fluctuation that assists the GJJ phase particle tunneling through the potential barrier $\Delta U$.

With 1-fW illumination, $\Gamma_{meas}$ is considerably higher than in the dark. At a constant photon flux, $\Gamma_{meas}$ increases monotonically with $I_b$. The absorbed photons can switch the GJJ more readily at a higher $I_b$

because the phase particle can escape over a lower $\Delta U$. Below ~ 2.7 $\mu A$, the junction may retrap before detection. This leads to false negative counts resulting in a reduction of quantum efficiency, $\eta$, defined as the number of measured single photons over the total number of photons absorbed into the detector. At ~ 2.7 $\mu A$, $\Gamma_{meas}$ starts to saturate, signifying that $\eta$ is approaching near unity, similar to superconducting nanowire detectors[7]. When $I_b > 3$ $\mu A$, the GJJ switches spontaneously by the MQT mechanism such that $\Gamma_{meas}$ is dominated by junction self-switching. The nonlinear $\Gamma_{meas}$ in the log-linear plot deviates from activation theory and underscores the detection of single photons as discrete events rather than a continuous heating[28]. Calibrated using $V_{refl}$ (see Supplementary Information), 1-fW photon illumination corresponds to 45 photons/s absorbed into the graphene. The right $y$-axis in Fig. 1D shows that $\eta \simeq 0.77 \pm 0.08$ when $\Gamma_{meas}$ saturates.

We can prove that each of the GJJ switching events is triggered by a single photon[8]. For a coherent state, the probability of an $m$ photon state, $\mathcal{P}_c(m)$, with a mean photon number, $\mu$, follows the Poisson distribution, i.e., $e^{-\mu}\mu^m/m!$. When $\mu \ll 1$, $\mathcal{P}_c(m=1)$ grows linearly with $\mu$, and hence the laser power. We measure the switching events of our detector over a range of laser powers for 300 seconds to obtain the switching rate, $\Gamma_{meas}$. We then calculate switching probability, $\mathcal{P} = \Gamma_{meas}/\mathcal{B}$, where $\mathcal{B}$ is the bandwidth of our detector upper bounded by the lowpass filters (30 kHz) used in biasing and measuring the GJJ[8,28]. Figure 1E shows that $\mathcal{P}$ depends linearly on laser power over several orders of magnitude, proving our detector is single-photon sensitive. Furthermore, Fig. 1E inset plots the distribution of $\Gamma_{meas}$. The histogram follows the Poisson statistics (solid lines) and the standard deviation constitutes the shot noise of uncorrelated photons from the coherent source.

**mK Optical Scanning**

To confirm the thermal detection of a single photon, we study the photon absorption by measuring $\Gamma_{meas}$ as the laser scans across the graphene. Figure 2A shows $V_{refl}$ as the beam rasters the transverse ($y$-) axis, 10 $\mu m$ away from the GJJ and parallel to the red dashed line in Fig. 1B. The data agrees well with the calculated spatial dependence of the $V_{refl}$ (solid line) by convolving a Gaussian profile of the 4-$\mu m$ beam

spot with a boxcar function representing the spatial extent of the graphene heterostructure (marked by the vertical dashed lines, see Supplementary Information). Specifically, the measured $V_{refl}$ also matches to our calculated ratio of the reflectance (horizontal dashed lines) of silicon to that of the graphene heterostructure. The excellent agreement supports that the calibration of the photon absorption into the monolayer graphene due to the interference effect from the graphene heterostructure is about 0.61% (see Supplementary Information). This can be improved up to 99% by a photonic cavity[29–31].

Figure 2B plots $\Gamma_{meas}(y)$ which resembles $V_{refl}(y)$, indicating that the absorbed photon switches the GJJ. When the beam spot is completely off the graphene, we measured zero $\Gamma_{meas}(|y|\geq 5\mu m)$, confirming that the stray light does not contribute to the measured single-photon counts. We normalize $\Gamma_{meas}(y)$ by the expected rate of absorbed photons to estimate $\eta$ (Fig. 2C, see Supplementary Information). Contrary to the variations of $\Gamma_{meas}(y)$ and $V_{refl}(y)$, $\eta(y)$ remains roughly a constant with an average value of ~0.8 when the beam spot illuminates the graphene.

## Benchmarking and Performance

To investigate the heat propagation from a single photon, we measured $\Gamma_{meas}$ in the longitudinal ($x$-) direction of the device (Fig. 1B purple dashed line). Figure 2D and E plots $V_{refl}(x)$ and $\Gamma_{meas}(x)$, respectively. By positioning the beam spot far away from the GJJ, we can ensure no clicks are due to Cooper pair breaking from photon exposure in the superconducting electrodes. Similar to $\Gamma_{meas}(y)$, $\Gamma_{meas}(x)$ subsides when the beam spot moves off the graphene absorber. Interestingly, $\Gamma_{meas}(x)$ remains high when the beam spot is positioned far away from the GJJ. By approximating our long flake as one-dimensional, we can understand this behavior using a dissipative diffusion equation[32]:

$$\frac{\partial}{\partial t} T_e^2 = \mathcal{D} \frac{\partial^2}{\partial x^2} T_e^2 - \frac{1}{\tau_{ep}} \left( T_e^\delta - T_0^\delta \right) \qquad (1)$$

with $\tau_{ep}$ being the decaying time constant of the E-Ph dissipation, $\delta$ being the E-Ph coupling power law, and $\mathcal{D}$ being the electronic diffusion constant which is given by $\sigma\mathcal{L}_0/\gamma_S$ where $\sigma$ and $\mathcal{L}_0$ are the electrical conductivity and Lorenz number, respectively. The first and second term on the right-hand side of Eqn. (1) represent the heat diffusion and dissipation, respectively. The ratio of these coefficients determines the characteristic length scale of heat diffusion, $l_D = \sqrt{\mathcal{D}\tau_{ep}} \simeq 230\,\mu m$ (see Supplementary Information), which is much longer than our sample length, leading to a small variation in $\Gamma_{meas}(x)$.

Figure 2F plots the $\eta(x)$. The suppression near both ends of the graphene are potentially due to the scattering of light by the metallic electrodes or, when the beam spot is near the GJJ, heat leakage directly into the superconductors when $k_B T_e$ exceeds $\Delta_s$ (~1.3 meV for our MoRe electrodes), or when the beam spot is far from the GJJ, due to E-Ph dissipation[32]. After accounting for the reduced area of graphene at the GJJ, $\eta(x)$ exhibits no noticeable variation as the beam spot approaches the GJJ. This suggests the GJJ switching mechanism is primarily governed by the bolometric effect[15,20], rather than quasiparticles[28,33] generated from the breaking of Cooper pairs when the superconducting electrodes of the GJJ are directly under photon illumination.

The performance of Dirac-fermion SPB depends on the electron density, $n_e$. Figure 3A shows $\Gamma_{meas}$ and $\eta$ vs. $I_b$ at various gate voltages, $V_{gate}$, with an absorbed photon rate of 45 Hz. As $V_{gate}$ decreases, $\Gamma_{meas}$ appears at lower $I_b$ because the GJJ critical current, $I_c$, is determined by $I_c R_n \propto \Delta_s$[23], where $R_n$ is the GJJ normal resistance. As shown in Fig. 3B, when $V_{gate}$ approaches the charge neutrality point at $-0.15$ V, the number of conduction channels decrease, and hence $I_s$, as a proxy for $I_c$, quenches with increasing $R_n$. The decreasing $I_s$ can degrade the GJJ

sensing in two ways: firstly, the reduced Josephson energy makes the GJJ susceptible to thermal noise, pushing the device from the MQT to TA regimes[34]; secondly, a smaller $\langle I_s \rangle - I_r$ value encourages the phase particle to retrap without the GJJ latching to the normal state. At $V_{gate} = 0.25$ V, $\Gamma_{meas}$ does not rise above the $\Gamma_{dark}$.

As $V_{gate}$ increases from 0.25 to ~4 V, $\Gamma_{meas}$ develops a plateau region near 45 Hz, regardless of $V_{gate}$ but corresponding to $\eta\gtrsim 0.8$ for $I_b > 0.85\langle I_s \rangle$, before the steep rise at the high $I_b$. This $\Gamma_{meas}$ plateau is the saturation of photon counting with a high $\eta$ shown in Fig. 1D. However, when $V_{gate}$ increases up to 7 V, $\Gamma_{meas}$ overlaps with $\Gamma_{dark}$ again. To better observe the performance of the SPB, we compare $\eta$ by normalizing $I_b$ to $\langle I_s \rangle$ at various $V_{gate}$. Figure 3C shows the evolution of the plateau and the optimal $V_{gate}$ (= 2.25 V) where the SPB enjoys simultaneously a high $\eta$ and low $\Gamma_{dark}$. For $V_{gate} < 0.5$ V, the sharp suppression of $\langle I_s \rangle$ leads to a reduction in $I_{bias}$ for which $\eta > 0.8$. We attribute this observation to $I_s$ approaching $I_r$, where the junction would have some probability of self-switching before latching could occur. For $V_{gate} \gtrsim 2.5$ V, $\langle I_s \rangle$ remains roughly a constant. Partially, we attribute the weakening of single-photon detection at higher $V_{gate}$ to the lower $T_{1p}$ due to a larger $c_e$ at higher $n_e(V_{gate})$. However, heat diffusion and thermal decay affect $\eta$ equally for all $V_{gate}$; $\sigma$, $\gamma_S$, and E-Ph coupling scale as $\sqrt{n_e}$, so the $n_e$ dependence cancels out in both $\mathcal{D}$ and $\tau_{ep}$[32]. In addition to the bolometric effect at high $V_{gate}$, we observe a curve in the log-$\Gamma_{dark}$ vs. $I_b$ plot that deviates from MQT or TA theory. This indicates additional noise inducing GJJ switching[35]. Better filtering and GJJ sensor design will prevent extra noise from eroding $\eta$ at high $V_{gate}$.

We can benchmark our Dirac-fermion SPB by exploring the competing tradespace between $\eta$ and $\Gamma_{dark}$[15]. At higher $I_b$, the GJJ can switch not only by the heat of a single photon, but also spontaneously by thermal or quantum fluctuations. Lowering $\Delta U$ with a higher $I_b$ can improve $\eta$, but at the cost of higher $\Gamma_{dark}$. Figure 3D plots the tradespace by extrapolating $\Gamma_{dark}$ from the MQT that is proven to dominate $\Gamma_{meas}$ in the absence of photons (Fig. 1D). $\eta$ grows with $\Gamma_{dark}$ as expected. At $V_{gate} = 2.25$ V, the device reaches $\eta \approx 0.87$ with $\Gamma_{dark}$ of 1 photon/s. At optimal $V_{gate}$, Fig. 1D shows $\eta \approx 0.75$ with $\Gamma_{dark}$ of 1 photon/week, corresponding to an ultralow effective NEP, $\epsilon_P\sqrt{\Gamma_{dark}}/\eta$[36], with $\epsilon_P$ as the photon energy, of $2 \times 10^{-22}$ W/$\sqrt{Hz}$. In the future, a kinetic inductance readout[37,38] can improve $\eta$ and increase the detector bandwidth, $\mathcal{B}$, while suppressing $\Gamma_{dark}$.

## Thermal Modeling

We can approximate the temperature rise of Dirac electrons by a single photon through the $\eta$ dependence on $I_b$. Since the thermal energy from a single photon needs to overcome $\Delta U$ to induce the escape of the phase particle, we obtain $\Delta U/k_B$ as a function of $I_b/\langle I_s \rangle$ and replot cuts from Fig. 3C in Fig. 3E. At $V_{gate} = 2.25$ V, the data suggests that a single photon can provide enough energy to overcome a $\Delta U/k_B$ of ~8 K, compatible with our estimation of $T_{1p}$ of ~2 K.

To gain more insight into the bolometric effect, we study $\eta$ versus $\Delta U/k_B$ at various $T_0$ and $V_{gate} = 2$ V. As shown in Fig. 4A, we are able to detect single photons up to 1.2 K, with a reduced $\eta$ of 0.5. When $T_0$ rises, $\eta$ reduces and the performance of our Dirac-fermion SPB degrades by several mechanisms: (1) the GJJ is subjected to more thermal noise, (2) the rise of $T_e$ from a single photon, $T_{1p}$, diminishes as $c_e$ increases, and (3) $\tau_{ep}$ shortens with a stronger E-Ph coupling. Between $T_0 = 0.02$ and 1.2 K, the Josephson plasma frequency remains much greater than $k_B T_0$ because $\langle I_s \rangle$ diminishes only by ~30%. Therefore, we neglect the temperature dependence of the GJJ and include only the bolometric effect in graphene, i.e., $c_e \propto T_0$ and $\tau_{ep} \propto T_0^{2-\delta}$, to model $\eta(T_0)$.

The single-photon enhanced escape probability of the phase particle out of $\Delta U$ can be approximated as $\eta(T_0) = 1 - \exp(-\Gamma_{1p}\tau_{ep})$ where $\Gamma_{1p}$ is the enhanced escape rate induced by a single photon that is proportional to $\exp(-\Delta U/k_B T_{1p})$ based on the activation theory of a thermal excitation $k_B T_{1p}$. Figure 4B plots the modeling result using

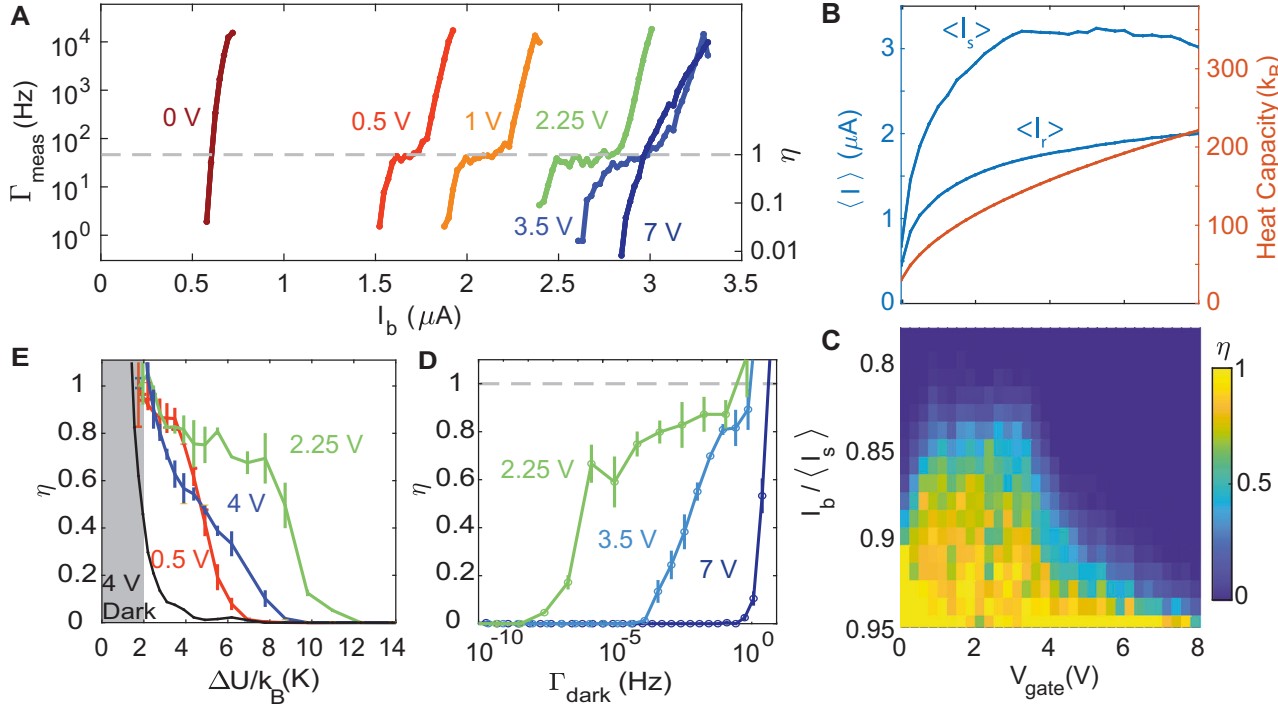

**Fig. 3 | Dependence of Dirac-fermion SPB performance on electron density.**
**A** Gate dependence of $\Gamma_{meas}$ and $\eta$ under 1 fW of laser power with the beam spot at position $(x, y) = (6, 0)$ $\mu$m. Charge neutrality is at $V_{gate} = -0.15$ V. **B** Gate dependence of measured $\langle I_s \rangle$ and $\langle I_r \rangle$ (blue), and calculated heat capacity of the graphene absorber (orange). **C** $\eta$ as a function of $V_{gate}$ and $I_b$. The region of high $\eta$ (yellow) under a relatively small $I_b$ marks the optimal performance of the SPB. **D** Tradespace between $\eta$ vs. dark count ($\Gamma_{dark}$) for three different $V_{gate}$. At an optimal $V_{gate}$ of 2.25

V, $\eta \approx 0.87$ (0.75) for a $\Gamma_{dark}$ on the order of 1 photon/s (1 photon/week). **E** $\eta$ vs. $\Delta U/k_B$ (for the Boltzmann constant $k_B$) of the washboard potential for three different $V_{gate}$. Gray box indicates the region where dominated by self-switching of the graphene Josephson Junction (GJJ). At $V_{gate} = 2.25$ V, a single photon can induce the escape of the GJJ phase particle from a $\Delta U/k_B$ of ~ 8 K. Error bars represent the standard deviation over multiple runs.

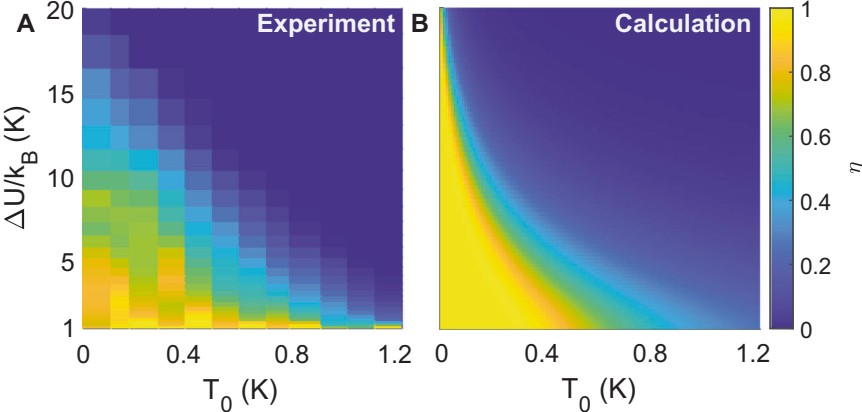

**Fig. 4 | Intrinsic quantum efficiency as a function of temperature and barrier height of the washboard potential. A** Experimental data. **B** Calculation using the temperature dependence of both electronic specific heat in graphene and electron-

phonon thermal decay time. Their qualitative agreement suggests a simple thermal model in graphene for describing our SPB.

$\delta = 4$ (E-Ph coupling in clean graphene, see Supplementary Information), at $T_0 = 20$ mK. We find that $\tau_{ep} = 75$ ns, and $T_{1p} = 2.5$ K best matches with the data in Figure 4A. Therefore, the three independently evaluated $T_{1p}$ from theory[15,32], measured $\Delta U$ (Fig. 3E), and thermal modeling (Fig. 4) are mutually consistent. The overall qualitative agreement between the spatial, electronic and thermal dependencies shown in this work demonstrates that a thermal model of Dirac fermions in graphene successfully describes our SPB.

## Methods

### Fabrication

Fabrication of our Dirac-fermion SPB begins with a high-resistivity silicon chip sputtered with 200 nm of niobium (Nb). Using photolithography and plasma etching, DC electrodes and a gate line are patterned from the Nb film. At the center of the pre-patterned Nb chip, a 200 $\mu$m-by-200 $\mu$m area of bare Si remains exposed for the placement of the graphene heterostructure. The hBN/graphene/hBN/

graphite heterostructures are prepared and placed using tape exfoliation and PDMS+chemical adhesive stacking techniques[39].

We use electron-beam lithography and plasma etching to define the heterostructure. The bottom graphite flake in the heterostructure serves as a gate to control the carrier density in the graphene, separated by the bottom hBN layer. This graphite layer also screens the graphene from charge inhomogeneities that may exist at the surface of the silicon. The graphite is connected using a MoRe electrode (75 nm thick), whose connection is severed from the graphene by plasma etching the top hBN and graphene.

In order to prevent short-circuiting between the graphene and graphite on either side of the heterostructure during the sputtering of the Josephson junction electrodes, both sides are insulated with a 120 nm poly(methyl methacrylate) (PMMA) layer. This layer is overdosed forming a cross-linked insulator. Afterward, the Josephson junction electrodes, made of MoRe (195 nm), are patterned by electron-beam lithography. For the primary device studied in this text, Device B, electrodes are sputtered onto a two-dimensional graphene sheet exposed by etching only the top hBN layer. For Device A, the electrodes are sputtered onto a one-dimensional graphene edge exposed by etching both the top hBN and graphene layers. Both Device A and Device B have a junction channel length of 600 nm and a width of 1.7 $\mu$m.

Lastly, electron-beam lithography is utilized to define the galvanic connection between the MoRe to the pre-patterned Nb film. To eliminate any oxide layer on the Nb and ensure superconducting contact, we employ in-situ argon ion milling before deposition. Without breaking vacuum, an adhesion layer of Ti (5 nm) is evaporated, followed by sputtering of MoRe (250 nm) on the freshly exposed surfaces. The process concludes with a lift-off in acetone to remove the excess metal.

## Laser reflectometry

We use laser reflectometry measurements (Supplementary Fig. 5) to accurately position the beam spot onto the Dirac-fermion SPB. Described in the main text, we use a single-mode optical fiber to bring 1550 nm photons to our samples through a long-pass filter. The fiber system successfully suppresses the stray ambient light from the laboratory space down to our SPB to merely 3 photons per minute. For reflectometry, the 0.6 (0.3) numerical aperture focusing lens for device B (A) is chosen to balance between the size of the beam spot and the collection efficiency of the light reflected from the samples back to the optical fiber. After reflecting off the device, the light is routed via a directional coupler to a PbTe photodetector. We modulate the incident light intensity by applying a sinusoidal voltage bias to the laser diode. We then measure $V_{\text{refl}}$ by a lock-in amplifier at different sample positions to produce the image in Fig. 1C. The directional coupler enables continuous monitoring of the incident laser power using a power meter.

Switching from laser reflectometry to single-photon measurements does not require any addition or removal of components in the optical path. We simply turn off the sinusoidal voltage bias to the diode, apply a small DC voltage bias to set the laser power output at 1 $\mu$W, and tune an in-line variable attenuator.

## $\Gamma_{\text{meas}}$ through sweeping and counting techniques

We measure $\Gamma_{\text{meas}}$ through two different, but equivalent, measurement protocols[40]. The first is to collect the GJJ switching statistics and extract $\Gamma_{\text{meas}}$ through the Fulton-Dunkelberger method[35]. In this protocol, we ramp $I_b$ from -4 $\mu$A to +4 $\mu$A and record the junction voltage, repeated over ~10$^4$ sweeps. For each sweep, we record $I_s$ at which the junction switches from superconducting to resistive. Collecting the statistics of $I_s$, we can extract a switching rate at each $I_b$[35].

Complementary to this approach is the counting method[40]. Here, we set a constant $I_b$ below $\langle I_s \rangle$ while monitoring the voltage across the

junction. When a switching event occurs, a voltage click is recorded. More specifically, the voltage from the device is referenced to a comparator and a DC voltage source. The DC voltage source is set well above the noise threshold of our digital to analog converter but below the normal state voltage. When a photon is absorbed, the junction switches, increasing the voltage triggering the comparator. The comparator sends a signal to a fast switch (response time much faster than the filter RC time) that sets $I_b < I_r$. The device then enters the superconducting state, where it has a lower voltage. Thus, it changes the signal of the comparator to switch the applied bias back to $I_b$. These two measurement techniques yield the same results[40].

## Description of the Calculation of $\eta$

In order to properly calculate $\eta$ we need a thorough understanding of three factors (1) The measured switching rate, $\Gamma_{\text{meas}}$, (2) the amount of photon flux being sent by the laser and (3) the amount of photon flux absorbed by the graphene, which can be further split between a geometric component and the absorption of the graphene within the van der Waals stack. The determination of $\Gamma_{\text{meas}}$ was addressed in the previous section.

For (2), to calibrate the power incident on the graphene, we use a combination of continuous power monitoring and measured optical attenuation at room temperature. We monitor the continuous optical power using the photodiode labeled "PM" in Supplementary Fig. 5 (ThorLabs S154C) and scaling it by the difference in attenuation between the optical path up to this photodiode (2 dB) and through the entire optical train (93 dB), 91 dB in total. These attenuation factors were measured at room temperature using the same photodiode, where the laser power is measured directly after the laser output, at the location of "PM" in Supplementary Fig. 5, and directly after the aspheric lens. Since the maximum output of the Fabry-Perot laser is 1.6 mW, the attenuated output power falls below the minimum resolvable optical power of the photodiode (10 pW). Therefore, to derive the total attenuation of the optical train, we remove the variable attenuator (ThorLabs VOA50-APC), separately characterize its attenuation (a maximum of 58.3 dB) and add this back to the attenuation of the optical train without the variable attenuator (35.7 dB). In principle, optical components that are cooled down will thermally contract and might exhibit additional insertion losses due to mechanisms like strain or misalignment. We measure this additional attenuation by measuring the laser reflectance signal of niobium at room temperature and at 20 mK. Assuming that the reflectance of niobium does not change when cooled down, there is no measurable difference between the reflectance signals at the two temperatures, implying that our optical train exhibits no additional attenuation when cooled down. Collectively, this process ensures a robust understanding of the photon flux reaching the mK stage of the fridge, $\mathcal{N}$.

The photon flux absorbed by the graphene contains a geometric, and a thin film interference component. The geometric component describes how much overlap the graphene region has with the beam spot. In the supplementary information we show that we are able to estimate the size of the beam spot by using our reflectometry measurement to image known device features on the chip. Here, we assume the beam spot adheres to a Gaussian beam profile, and we measure $V_{\text{refl}}$ as the beam spot passes over a known feature. The image formed from $V_{\text{refl}}$ is the convolution of the Gaussian beam spot and the feature. From here we are able to extract the beam waist and subsequently the proportion of the beam spot which overlaps with the graphene, $\alpha_{\text{geo}}$ (see Supplementary for more details).

The graphene absorption is found using the wave-transfer matrix method. This method accounts for the index of refraction of each layer of the van der Waals heterostructure and the subsequent reflection/ transmission/absorption at each interface to extract the absorption into the graphene layer. In our devices, we find a graphene absorption

$\alpha_{gr} \approx 0.6\%$ (see Supplementary for detailed calculation and relevant layer thicknesses).

In order to calculate the number of photons absorbed by the graphene we thus take $\mathcal{N}\alpha_{\text{geo}}\alpha_{gr}$. From here, $\eta = \Gamma_{\text{meas}}/\mathcal{N}\alpha_{\text{geo}}\alpha_{gr}$.

## Data availability

All data and codes used to produce the main text figures are available online at on Zenodo at https://doi.org/10.5281/zenodo.18497494.

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

## Acknowledgements

We thank J.Balgley and B.-I.Wu for valuable discussions, and B.Hassick for technical support and design. We further thank J. Balgley for rendering Fig. 1A. Experimental setup, measurements and data analysis performed by B.H. were supported by an appointment to the Intelligence Community Postdoctoral Research Fellowship Program at the Massachusetts Institute of Technology, administered by Oak Ridge Institute for Science and Education through an interagency agreement between the U.S. Department of Energy and the Office of the Director of National Intelligence, and by E.G.A. who was supported from the Army Research Office MURI (Ab-Initio Solid-State Quantum Materials) Grant no. W911NF-18-1-043. Sample optimization, characterization, and fabrication by W.J. and G.-H.L. were supported by National Research Foundation of Korea (NRF) funded by the Korean Government (RS-2022-NR068223, RS-2024-00393599, RS-2024-00442710, RS-2024-00444725, RS-2025-02317602), ITRC program (IITP-2022-RS-2022-00164799) funded by the Ministry of Science and ICT, Samsung Science and Technology Foundation (SSTF-BA2101-06, SSTF-

BA2401-03), Samsung Electronics Co., Ltd (IO201207-07801-01), and Agency for Defense Development funded by Defense Acquisition Program Administration (DAPA) (UI257011TE*) For hBN, K.W. and T.T. acknowledge support from the JSPS KAKENHI (Grant Numbers 21H05233 and 23H02052) and World Premier International Research Center Initiative (WPI), MEXT, Japan. Modeling was performed by C.F., B.H., E.G.A., and K.C.F. Data analysis and manuscript preparation were performed by B.H., E.G.A., W.J., B.J.R. D.E., G.H.L, K.C.F., and E.A.H. who acknowledges support under NSF CAREER DMR-1945278.

## Author contributions

B.H. and E.G.A performed the experimental setup, measurements and data analysis. W.J. and G.-H.L. performed the sample optimization, characterization, and fabrication. K.W. and T.T. provided the hBN. C.F., B.H., E.G.A., and K.C.F performed the modeling. B.H., E.G.A., W.J. B.J.R., D.E., G.H.L, K.C.F., and E.A.H. performed the data analysis and contributed to manuscript preparation.

## Competing interests

The authors declare no competing interests.
