## [Transparent Peer Review file · Nature Communications]

Thermal Detection of Single Photons Using Dirac Fermions

Corresponding Author: Professor Kin Chung Fong

Version 0:

Reviewer comments:

Reviewer #1

(Remarks to the Author)

In this work, the authors experimentally demonstrated a graphene-Josephson weak link bolometer single photon detector. Electron gas in graphene is used as a photon absorber, and the Josephson weak link is biased close to switching. Upon single photon absorption, the electron gas in graphene is heated up, which results in the switching of the Josephson weak link. Once switched, the weak link remains in resistive state, allowing the photon event to be registered. The device is then reset for the next photon event. The authors demonstrate quantum efficiency up to ~ 0.8 , and a very small NEP suggesting excellent sensitivity above the dark counts.

Overall, the research is solid and significant, and the manuscript is reasonably clear written. I have several questions before judging the suitability for publication in Nature Communications.

- 1) The novelty of this work and comparison with other single photon detector concepts. There are many commercially developed and new proof-of-principle NIR single photon detectors. Can the authors briefly compare this work with other types of NIR single photon detectors: what are the demonstrated strengths and/or weaknesses of the devices studied in this work, compared to the current state-of-the-art?
- 2) A key step in the operational procedure of the devices discussed here is to reset the weak link to the superconducting state after it is switched to resistive state by a photon. What is the reset speed? It seems that the reset time affects the maximum photon flux. Can the authors comment on the range of photon flux that can be reliably measured?
- 3) Can the authors elaborate a bit more on the switching mechanism in the devices studied here? In particular, how does the electron temperature rise in the normal channel trigger a switching while the superconducting leads (sources of the Cooper pairs and macroscopic phases), presumably, remain at low temperature?
- 4) In the introduction, the authors state that an advantage of graphene absorber is the extremely low electronic specific heat at the charge neutrality point. However, later in the discussion, the measurements are mainly in the mid-doping regime which has a sizable supercurrent. This feels a bit confusing. Does it mean that the current device scheme cannot take advantage of the extremely small electronic heat capacity at the charge neutrality?
- 5) It helps to provide more details on the graphene/hBN stack. For example, what are the thicknesses of the hBN layers? Such information helps the readers to judge the relation between gate voltage and doping. It also helps to clearly specify the carrier density at the gate voltages.
- 6) The temperature increase of $\sim 2\text{K}$ upon absorption of a single photon is very large compared to the base temperature. When the authors estimate parameters such as E-Ph scattering time, diffusion constant, etc, is this large temperature increase considered?
- 7) Can the authors comment on the impact of photon energy and detection speed on the performance of the SPB? In particular, the very small NEP is largely due to the small dark count rate, resulting from the bias current being significantly lower than the switching current. Increasing the sampling bandwidth may increase the noise of the circuit; and for lower energy photons, the bias current may need to be closer to the switching current for decent efficiency. What is the optimized photon energy and flux for the devices studied in this work? Are there significant challenges for high-speed counting of low energy (e.g., THz) photons?
- 8) In figure 3A, the bias-efficiency dependence for large gate voltage overlaps with the dark counts. The authors have some discussions in the manuscript on this observation. However, I still don't quite understand the reasons for such degradation in performance. Can the authors comment on this and clarify its discussion?

Reviewer #2

(Remarks to the Author)

The authors present interesting and relevant analysis of graphene as a bolometer. The measurement of low dark counts, and proof of single photon detection are particularly significant.

- The authors base the calculation for the internal quantum efficiency, η , on an estimation of the power incident on the graphene. The absorption of the graphene layer is well evidenced to be 0.6% based on the wave-transfer matrix, but it is not clear how the power incident on the device is calibrated. Specifically, the losses in the focusing lens assembly, long pass filter, the efficiency of the directional coupler, and losses in the fiber, etc, are assumed but not measured. Can the authors provide further information on how these assumptions were made, and provide any detail further detail on whether any of these losses were measured, and how it affects the uncertainty on η ?
- In the introduction, the authors motivate the paper by saying “the energy gap provides a mechanism to distinguish photons from dark counts caused by fluctuations, but also limits the detection of lower-energy photons”. Superconducting detectors have established single photon detection far into the mid-infrared, out to 30 μm and beyond. This paper only demonstrates detection of 1550 nm photons. Can the authors comment on the low-energy limit of photons they believe this graphene-based device can detect?
- The authors say on line 95 that they “register a click, reset the bias current” Can the authors further expand upon how this bias current is reset? Is it manually switched off upon detection? Is an inductive shunt or alternative low-resistance path to ground used to naturally divert the current?
- The authors have a very in depth description the calculation of η in the supplement, but very little description of how it is calculated in the main text. Adding further detail of how η is calculated to the main text would be beneficial.
- Line 171 “photon” should be “photons”
- In Fig 2D, the V_{refl} peaks on the right side of the chart. It would be useful to denote in the chart itself that this is due to reflection off of the contact. Additionally, it would be useful to add a gray dashed line indication the device, to mirror the line in 2A, given that the vertical scale bars are different.
- Line 232: “corresponding to $\eta \geq 0.8$ over I_b ranges $\sim 15\%$ of $\langle I_s \rangle$,” This sentence is unclear, do they mean within 15% of I_s ? If so, it may be more clear to say something along the lines of $>85\%$ of I_s .
- The authors don’t show any data of I_b for negative V_{Gate} . Is the photon detection behavior symmetric about the charge neutrality point?
- In the Table S2, the authors seem to use α_0 to denote the freestanding reflectance, but I can’t find any place where this is explicitly stated.
- In Figure 3B, how is I_r as a function of V_{gate} determined?
- On Line 240, the authors say “For $V_{\text{gate}} < 0.5\text{V}$, the sharp suppression of I_s leads to a poor η .” The behavior of the device in this regime is unclear, and this sentence should be expanded upon qualitatively for clarity. Why doesn’t it work near the charge neutrality point?
- The thicknesses of the graphene and hBN layers determine the reflectances calculated in the wave-transfer matrix. Table S2 lists the thicknesses of the materials, but it is unclear how these thicknesses are measured. Specifically the bottom graphite will have a large impact on the reflectance of the device, and it is unclear how confident they authors are they have 1.33 and 2 nm thickness for d4.

Reviewer #3

(Remarks to the Author)

Review of the manuscript by B. Huang et al., “Thermal Detection of Single Photons Using Dirac Fermions”

The manuscript reports on the development of a single infrared photon counter based on a graphene absorber integrated with a graphene-based Josephson junction. The authors’ central idea is to spatially separate the photon absorption site—where the absorption event raises the electron temperature in graphene—from the detection site, where the resulting temperature rise is sensed via the switching of the Josephson junction into the resistive state.

The small heat capacity of graphene at low carrier density, close to the Dirac point, enables a pronounced temperature increase upon photon absorption, which in turn enhances device sensitivity. The spatial separation between the absorber and the Josephson junction also mitigates detrimental effects of high-energy photons on the superconducting properties of the junction leads.

The experimental results convincingly support the proposed detection mechanism. The authors present clear evidence that the device response peaks at the Dirac point and systematically evaluate the response while scanning the photon beam spot across the graphene absorber. These measurements are complemented by a solid theoretical analysis of heat propagation along the graphene strip, yielding a coherent and consistent picture of the device operation.

However, I see at least two major points for improvement of the manuscript in its current version, these are: a) Lack of performance benchmarking, and b) 2. Clarification of application scope – these points are elaborated below:

a. Lack of performance benchmarking

The manuscript does not compare the demonstrated infrared photon counter with the state-of-the-art performance of similar devices. Given that near-infrared single-photon detectors are a key enabling technology for quantum information science, the field is highly competitive. Several relevant works—absent from the current reference list—should be cited and discussed, including:

o R. E. Warburton et al., Optics Letters 32, 2266 (2007)
o W. Guo et al., Applied Physics Letters 110, 212601 (2017)
o I. Craiciu et al., Optica 10, 183 (2023)

A quantitative comparison of key performance metrics (efficiency, timing jitter, dark count rate) with these and other leading technologies would allow readers to assess the novelty and competitiveness of the reported device.

b. Clarification of application scope

The manuscript briefly mentions potential applications to axion searches. However, the photon energy range relevant to such experiments (10^{-6} – 10^{-3} eV, corresponding to frequencies below ~ 0.25 THz) is far below the ~ 200 THz photon energies studied here. This significant mismatch should be explicitly acknowledged. (Here I refer to a comprehensive review by Bradley, et al., ‘Microwave cavity searches for dark-matter axions’, Rev. Mod. Phys 75, 777 (2003))

In this context, the authors should also reference recent work directly targeting the relevant frequency range, such as:
o L. Pankratov et al., Nature Communications 16, 3457 (2025), which reports single-mode thermal microwave photon detection at 30 GHz using a similar superconducting sensor and currently sets the state-of-the-art in that domain.

Summarizing, the manuscript presents a technically sound and carefully executed study, supported by both experimental and theoretical analysis. The work is of interest to the community working on graphene-based superconducting devices and photon detection. However, in its current form, it does not convincingly demonstrate performance beyond the state-of-the-art, nor does it provide a sufficiently detailed comparison with competing technologies.

Given these limitations, I do not recommend publication in Nature Communications at this stage. A strengthened manuscript should include a thorough benchmarking against existing devices and a clearer discussion of realistic application domains.

Version 1:

Reviewer comments:

Reviewer #1

(Remarks to the Author)

The authors gave satisfactory replies to my questions and I feel that the revised manuscript is ok to be published at Nature Comm.

Reviewer #2

(Remarks to the Author)

Thank you for the clarifications in this review, the clarity of the paper is undoubtedly improved, and all of my questions have been answered. My only remaining comment is that in the in the new description of the calculation of eta in the Methods section, the location of “PM” is referenced as being in Fig S7 once and Fig S5 another time, I believe both should be S5 (lines 609 and 616).

Reviewer #3

(Remarks to the Author)

I would like to thank the authors for the clarifications provided in the revised version of their manuscript.

Reviewers' reports in black. Authors' response in blue.

Changes in the manuscripts highlighted in red in the redlined version in the submitted pdf (separate file).

Reviewer #1 (Remarks to the Author):

In this work, the authors experimentally demonstrated a graphene-Josephson weak link bolometer single photon detector. Electron gas in graphene is used as a photon absorber, and the Josephson weak link is biased close to switching. Upon single photon absorption, the electron gas in graphene is heated up, which results in the switching of the Josephson weak link. Once switched, the weak link remains in resistive state, allowing the photon event to be registered. The device is then reset for the next photon event. The authors demonstrate quantum efficiency up to ~ 0.8 , and a very small NEP suggesting excellent sensitivity above the dark counts.

Overall, the research is solid and significant, and the manuscript is reasonably clear written. I have several questions before judging the suitability for publication in Nature Communications.

1) The novelty of this work and comparison with other single photon detector concepts. There are many commercially developed and new proof-of-principle NIR single photon detectors. Can the authors briefly compare this work with other types of NIR single photon detectors: what are the demonstrated strengths and/or weaknesses of the devices studied in this work, compared to the current state-of-the-art?

We thank the reviewer for this comment and we agree that we should place our work into the larger context of state-of-the-art single photon detectors. We have modified the last sentence in the 2nd paragraph to highlight the key conclusion with respect to the comparison that we have added in the supplementary information in a table format. While the table compares our result to several more conventional platforms (SNSPD, mKID, SPAD). We choose to emphasize a body of SNSPD results for comparison because we believe this is the most comparable technology to our detector. Specifically, even though our result is only a proof-of-principle experiment, we find that our efficiency at a given dark count rate is comparable to state-of-the-art SNSPDs. This low dark count rate is due to the exponential suppression of dark counts due to the potential of the phase particle of the Josephson junction, which represents a systematic strength of our approach. While our timing jitter is comparable to SNSPD's (see C. Fried, et. al. Phys. Rev. Appl., **21**, 014006, [2024]) in theory, this value was not demonstrated in this work. Our count rate is presently a weakness as we are using DC switching of Josephson junction in this proof-of-concept experiment. However, we would like to emphasize that this is extrinsic, due to the external circuitry (as elaborated

on in our next response). We can improve the count rate up to ~ 100 Mc/s due to a combination of the intrinsic limit from the thermal time constant, and the extrinsic limit from the resonator readout technique. A more crucial weakness of our device is the fabrication complexity, relying on presently unscalable van der Waals stacking methodologies. We hope our proof-of-concept experiment can inspire more efforts in scalable fabrication, given the important application of single-photon detectors. Overall, we believe these metrics and theoretical limits demonstrate that this is an exciting platform, with many complementary avenues with the existing state of the art single photon detectors.

TABLE S6. Comparison of Detector Platforms. Efficiencies marked with a *, denote system efficiency.

Work	Platform	Efficiency	DCR	Timing Jitter	Count Rate
This Work	GJJSPD	87%	1/s	2.7 ps [46]	1.5 kc/s
Ref. [68]	SNSPD	78%	158/s	50 ps	1.5 Gc/s
Ref. [69]	SNSPD	1%	100/s	68 ps	100 Mc/s
Ref. [70]	SNSPD	28%	6×10^{-6} /s	—	—
Ref. [71]	SNSPD	93%*	1/s	150 ps	25 Mc/s
Ref. [72]	mKID	0.38%	—	—	—
Ref. [73]	SPAD	50%*	20000/s	70 ps	1 Mc/s

2) A key step in the operational procedure of the devices discussed here is to reset the weak link to the superconducting state after it is switched to resistive state by a photon. What is the reset speed? It seems that the reset time affects the maximum photon flux. Can the authors comment on the range of photon flux that can be reliably measured?

We thank the reviewer for this comment and have added the following discussion to the supplementary information, with the below figure:

To answer the question of reset speed, we consider both the rise and fall time of the voltage across the Josephson junction. The rise time of the device is effectively “instantaneous” because the switching of Josephson junction from supercurrent to resistive state increases sharply as a voltage step. On the other hand, the reset is limited by the cryogenic filters which are designed at a 30 kHz cut off frequency (see Figure Panel A for a time domain trace of a single photon switching event). These filters are external to the chip and are mounted directly onto the plates of the dilution refrigerator. Therefore, in this work, the measurement bandwidth is limited to be under 30 kHz.

Similarly, care must be taken to ensure that the absorbed photon flux rate is well below the filter's roll-off. As an example, the highest powers presented (~ 1 fW) corresponds to a flux of ~ 7800 photons/s, and (with an absorption of 0.6%) ~ 50 absorbed photons/s placing the presented powers well under the cut-off frequency of the filter. To benchmark the maximum flux rate we measure Γ_{meas} at higher photon fluxes (see Figure Panel B). Here, powers as high as 30 fW (1500 photons/s absorbed by graphene) still show a single photon detection without substantial correlation effects from discharge (as evidenced by the lack of change of biases that show self switching, i.e the high I_b regime). However, for 100 fW (5000 photons/s absorbed by graphene) the curves move towards lower values of I_b suggesting either the device itself has begun overheating or correlation effects from the RC discharge are becoming non-negligible. Given that the absorbed photon flux rate is approximately an order of magnitude less than the filter discharge rate (30 kHz), we ascribe this feature to correlation effects from heating.

The device itself can likely support much faster reset times with a higher-cutoff frequency filter and low stray capacitance by careful engineering. The thermal time constant has been measured to be ~ 10 ns (see Lee Nature, 586 2020), which can serve as a lower bound of the reset time. To unlock the full potential, we would need to embed the junction within a microwave resonator circuit, e.g. Katti et al NanoLett 23, 10 (2023), for continuous operation of microwave bolometry. In this case, the response time would be limited to the bandwidth of the resonator readout or the thermal time constant (whichever is longer).” We are currently working on a model for this resonator readout. We thank the reviewer for this insightful question.

3) Can the authors elaborate a bit more on the switching mechanism in the devices studied here? In particular, how does the electron temperature rise in the normal channel trigger a switching while the superconducting leads (sources of the Cooper pairs and macroscopic phases), presumably, remain at low temperature?

We thank the reviewer for this question. The switching mechanism and the modeling are discussed in Walsh et. al. Phys Rev Appl, **8**, 024022 (2017) (Ref. [15]) and C. Fried, et. al. Phys. Rev. Appl., **21**, 014006, (2024) (Ref. [33]), and more recently in a more detailed materials/device design study (Jung et. al. arxiv 2503:86050 [2025]).

In short, the rise of electron temperature in the normal channel can suppress the critical current of the Josephson junction. This can lead to an enhanced switching rate while the superconducting leads remain at low temperature, as the Reviewer correctly pointed out. In greater detail and response to the Reviewer's question, we have added a description in the supplement for clarity. This reads:

"Microscopic understanding of the switching mechanism:

In superconducting-normal-superconducting Josephson junctions, supercurrent is mediated via Andreev reflection. Here, an incoming electron retroreflects off of a superconducting electrode as a hole (and vice versa). In a Josephson junction, repeated Andreev reflections between electrons and holes generate Andreev bound states – an energy spectrum which is dependent on the weak-link material, superconducting gap and phase.

The supercurrent mediated by the Andreev spectrum is $\sum_i dE_i/d\phi_i * f(E)$, i.e. the sum of the slopes of the occupied Andreev bands. At low temperatures, the Andreev spectrum allows for the maximum measured supercurrent through the device, however as temperature rises, the electron-hole symmetry of the spectrum allows for states with opposite current contributions to become occupied, thus lowering the critical current mediated by the junction.

In the case of the graphene SPD, an incoming photon is absorbed by the graphene electrons, causing an increase in temperature that propagates towards the junction (see C. Fried, et. al. Phys. Rev. Appl., **21**, 014006, [2024]), while the superconducting leads remain at the bath temperature. The leads are largely unaffected due to their sizable superconducting gap and strong thermal anchoring, whereas the hot electrons in graphene broaden the distribution that feeds the ABSs. This temperature rise increases the occupation of states above the Fermi level, thereby reducing the critical current.

Fundamentally, the Andreev spectrum enables this work in two ways: 1) The Andreev spectrum necessitates that the critical current is dependent on the graphene electronic temperature, enabling the mapping from the microscopic of the junction to the macroscopic circuit parameters (i.e. the change in critical current)" 2) The semi-continuous Andreev spectrum reduces the energy scaling from the superconducting gap, to the lower energy scale of the Andreev level spacing. In the limit of many modes this is nearly continuous and allows us to unlock the zero-bandgap of the graphene electrons."

4) In the introduction, the authors state that an advantage of graphene absorber is the extremely low electronic specific heat at the charge neutrality point. However, later in the discussion, the measurements are mainly in the mid-doping regime which has a sizable supercurrent. This feels a bit confusing. Does it mean that the current device scheme cannot take advantage of the extremely small electronic heat capacity at the charge neutrality?

We thank the reviewer for this good question. Indeed we cannot take full advantage of the low heat capacity **exactly on** the Dirac peak because the functioning of the Josephson junction as the thermal sensor also depends on the carrier density in graphene. Yet, in all gate voltage ranges, we are operating with exceedingly low electron densities between 10^{11} - 10^{12} cm^{-2} . Furthermore, the weak electron-phonon coupling also adds an advantage to graphene SPB. When compared with other bolometric materials, graphene single-photon bolometer consists only of Dirac fermions, without phonons. Typically the heat capacity in graphene electrons is on the order of $1 k_B$ per μm^2 . Since both the low heat capacity and the decoupling of electrons and phonons are derived from the Dirac fermions, we introduce the physics to a broader audience in this way in the introduction. We commend Reviewer's critical question to go a step further into this fascinating subject and application.

Before we go into the details further, we would like to make a correction and clarification. In preparing the response to this question, we realized a labeling error in Figure 3A. The left-most curve was labeled as "0.25 V", when in fact it should read "0 V". The curves for 0 V and 0.25 V are plotted below. We can in fact see the single photon detection at 0.25 V, albeit in a limited bias range. This corresponds to a density of $1.34 \times 10^{11}/\text{cm}^2$, which is near but not exactly on the Dirac point, and correspondingly low C_e of $\sim 49 k_B$.

Near the Dirac point (for instance at 0 V), the hysteresis of the junction is very small. The switching and retrapping currents become too close together to resolve a single photon. This is because even if the junction may switch to normal upon absorbing a photon, the device retraps instead of latching (within the response time of the plasma frequency of the junction). This is evidenced by the reduction in the single photon detection region's extent in I_{bias} as the density approaches the Dirac peak. This problem could potentially be resolved by increasing the width of the Josephson junction, which would increase the critical current of the device, however this would come at the cost of increasing the heat capacity due to the increased graphene area.

Further, near the Dirac peak, the low density of graphene electrons near charge neutrality often leads to charge puddles (see Samaddar et. al. PRL **116**, 126804 [2016]), which can lead to disruptions in electron flow and enhanced cooling rates. While care was taken (i.e. by using hBN substrates and graphite backgates for screening), it is impossible to rule these out in our device.

5) It helps to provide more details on the graphene/hBN stack. For example, what are the thicknesses of the hBN layers? Such information helps the readers to judge the relation between gate voltage and doping. It also helps to clearly specify the carrier density at the gate voltages.

We thank the reviewer for this comment. The thicknesses of the hBN layers were determined by AFM, as described in detail in our response to Reviewer 2 (Question 12). For clarity, we summarize the relevant information here.

In our devices, the carrier density was extracted using a capacitance model, with the bottom graphite serving as the gate electrode. The key parameters are provided in Table S1. Specifically:

- **Device A:** bottom hBN thickness ≈ 56 nm; carrier density per gate voltage $n_e/V_{\text{gate}}=0.22 \times 10^{12} \text{ cm}^{-2} / \text{V}$.
- **Device B:** bottom hBN thickness ≈ 36 nm; carrier density per gate voltage $n_e/V_{\text{gate}}=0.34 \times 10^{12} \text{ cm}^{-2} / \text{V}$.

These values allow a direct assessment of the relation between gate voltage and doping level in our devices. For completeness, the full list of parameters for all measured devices is included in Table S1. For clarity we have added a pointer to that table in the main text:

“we use a graphene-based Josephson junction (GJJ) (fabrication details in Methods, and a table of all relevant device parameters, **such as junction dimensions and flake thickness can be found in Supplementary Table 1)**”

6) The temperature increase of $\sim 2\text{K}$ upon absorption of a single photon is very large compared to the base temperature. When the authors estimate parameters such as E-Ph scattering time, diffusion constant, etc, is this large temperature increase considered?

Thank you for this thoughtful question from the reviewer. Yes, we have included the estimation of the E-Ph scattering time, diffusion constant, etc. to understand the large temperature rise, i.e. 2 K from a single NIR photon. In particular we have previously published these calculations in Ref. (see C. Fried, et. al. Phys. Rev. Appl., **21**, 014006, [2024]). Specifically, in the limit of a E-Ph scattering time longer than the diffusion time scale, the temperature rise is determined by the electron heat capacity of the total area of the graphene flake. This is expected because if the heat leakage out into the phonon from the graphene electrons is small, the temperature rise can reach a peak value uniformly across the entire graphene flake. As the energy is conserved within the graphene electron system, the temperature rise shall be given by the photon energy divided by the total electron heat capacity. In (Fried, et. al. Phys. Rev. Appl., **21**, 014006, [2024]) we show at the time scale for E-Ph scattering in clean graphene is of order 10 us.

Based on the electrical transport measurement, we can estimate the electron mobility of the graphene ($\sim 9100 \text{ cm}^2/\text{Vs}$, see supplementary table S1). The diffusion constant is $D = k_B T/e \cdot (\text{mobility})$. At 2K, $D = 1.57 \text{ cm}^2/\text{s}$. The characteristic diffusion time is then $t_{\text{diff}} = L^2/(4D)$, where $L \sim 10 \mu\text{m}$ is the length of the device. This gives $t_{\text{diff}} = 320 \text{ ns} \ll 10 \mu\text{s}$. Therefore we can justify that E-Ph scattering time is longer than the diffusion time scale in our samples at the lowest temperature, 0.02 K, we performed our experiment.

Beyond this, in the thermal modeling performed in Figure 4, we consider the photon-induced activation theory of the Josephson junction as the bath temperature is increased. While this calculation is more at the device level, the materials level considerations are still at play. As mentioned in the supplementary information section K, we take the integration time of the device to be the thermal time constant due to E-Ph scattering. This is taken as

$\tau_{ep} \propto \tau_{ep}(T_o = 20 \text{ mK})T_o^{2-\delta}$ where T_o is the bath temperature. It is important to note, while

the electronic temperature rise of ~ 2 K is large compared to the bath temperature, it is minute compared to the Bloch-Gruneisen temperature (~ 100 K for $n_e \sim 10^{12}/\text{cm}^2$).

Therefore, even in the event that the entirety of the heat was transferred into the graphene phonons, the phonon momentum would be well below the size of the Fermi surface and the electron-phonon coupling would be negligible. As such, the graphene electrons and phonons are out-of-equilibrium during the photon detection. We have previously published detailed calculations considering the variation of these parameters in the thermal modeling in Ref. [C. Fried, et. al. Phys. Rev. Appl., 21, 014006, (2024)].

7) Can the authors comment on the impact of photon energy and detection speed on the performance of the SPB? In particular, the very small NEP is largely due to the small dark count rate, resulting from the bias current being significantly lower than the switching current. Increasing the sampling bandwidth may increase the noise of the circuit; and for lower energy photons, the bias current may need to be closer to the switching current for decent efficiency. What is the optimized photon energy and flux for the devices studied in this work? Are there significant challenges for high-speed counting of low energy (e.g., THz) photons?

We agree with the reviewer that *for lower energy photons, the bias current may need to be closer to the switching current for decent efficiency*. Specifically, we expect, as the reviewer suggested, smaller photon energies can worsen the dark count for a given quantum efficiency and we have confirmed this effect from our model of MIR, FIR, and microwave photon detection: Ref. [C. Fried, et. al. Phys. Rev. Appl., 21, 014006, (2024)] Fig. 7c and Walsh et. al. Phys Rev Appl, 8, 024022 [2017] Fig. 7, which are consistent to our experimental result in the NIR result (Fig. 3D in maintext). The goal of our modeling and materials design works (Jung et. al. arxiv 2503:86050 [2025]) is to optimize this exact trade off by engineering the graphene to maximize the temperature rise of a given input photon energy and therefore increase the amount of signal from the photon. The detection speed is unchanged because our SPB in this work is extrinsically limited by the DC measurement.

Within the context of just this work, we can find the expected efficiency of detection vs bias by extending our thermal modeling from Figure 4b. As a reminder, here we considered the efficiency in which a 1550 nm single photon could overcome a potential barrier ΔU as a

function of bath temperature, T_b . If we instead keep $T_b = 20$ mK, we can scale the effective temperature rise, T_{1p} of the graphene electrons due to the absorption of a single photon. For Fig 4b we found good agreement for $T_{1p} = 2.5$ K @ $f_{\text{phot}} \sim 193$ THz (1550 nm). T_{1p} scales as the $f_{\text{phot}}^{-0.5}$. By varying this frequency we can produce the below plot of the expected η as a function of photon wavelength. We see a reasonably robust η of near unity for $\Delta U/k_B$ of ~ 4 K into the mid-IR, which is consistent with our thermal modeling in [C. Fried, et. al. Phys. Rev. Appl., 21, 014006, (2024)]. This is to show that we have a design strategy that is well-supported by modelling and calculations such that the graphene SPB can maintain a reasonably good performance, comparable to the result in this manuscript. We can bias the graphene SPB accordingly to detect lower energy photons with this sizable temperature rise.

We agree with the Reviewer that increasing the sampling bandwidth may increase the noise of the circuit. Filters must be carefully designed and incorporated as sampling bandwidth increases. More engineering efforts are needed in the future.

To achieve high-speed counting of THz photons, we will need to use a resonator readout. The challenge is to incorporate simultaneously the resonator readout and antenna of THz photons. We hope we will be able to solve this problem and demonstrate the graphene SPB in the application in THz if there is funding opportunity.

8) In figure 3A, the bias-efficiency dependence for large gate voltage overlaps with the dark counts. The authors have some discussions in the manuscript on this observation. However, I still don't quite understand the reasons for such degradation in performance. Can the authors comment on this and clarify its discussion?

We thank the reviewer for this comment as it is important to understand the performance of our SPD. The overlap of the data at $V_{\text{gate}} = 3.5$ and the dark count rate at $V_{\text{gate}} = 7$ V in Fig. 3A is in part due to the small, and non-monotonic change of the switching currents at gate voltages larger than about 3 V (Fig. 3B).

More generally, we should expect a degradation in the performance of our SPD when the gate is tuned far away from the Dirac peak. This is because, while critical current roughly saturates, the electronic heat capacity continues to grow, reducing the maximum temperature rise and thus our ability to detect single photons.

In addition to the above two points, we speculate that at higher gate voltages our Device B becomes impedance matched with an unused microwave circuit galvanically connected to the graphene junction which increases the noise of our detector (this effect was not present in Device A despite being fabricated on a nearly identical superconducting chip). This is evidenced by the failure of our MQT fits at elevated gate voltages (see figure from Fig S11 below). Here, we notice an increase in the junction resistance as a function of increasing gate voltage for the best fits. Simultaneously, the fits become less effective at describing the data. Further, at $V_{\text{gate}} = 7$ V we see a very slight “kink” in the data near $I_b = 3$ μA . This feature was observed in [Fulton, Dunkelberger PRB 9, 11 (1974)] and ascribed to extrinsic noise.

In the maintext, we believe that it is important to show the full range of gate values measured. The values reported and highlighted in this work are representative of the platform (as evidenced by the strong MQT fits at lower V_{gate} and the lack of this extrinsic noise in Device A).

Reviewer #2 (Remarks to the Author):

The authors present interesting and relevant analysis of graphene as a bolometer. The measurement of low dark counts, and proof of single photon detection are particularly significant.

- 1) The authors base the calculation for the internal quantum efficiency, η , on an estimation of the power incident on the graphene. The absorption of the graphene layer is well evidenced to be 0.6% based on the wave-transfer matrix, but it is not clear how the power incident on the device is calibrated. Specifically, the losses in the focusing lens assembly, long pass filter, the efficiency of the directional coupler, and losses in the fiber, etc, are assumed but not measured. Can the authors provide further information on how these assumptions were made, and provide any detail further detail on whether any of these losses were measured, and how it affects the uncertainty on η ?

We thank the reviewer for the question – this is an important detail that has been added to the Methods section of our paper.

“To calibrate the power incident on the graphene, we use a combination of continuous power monitoring and measured optical attenuation at room temperature. We monitor the continuous optical power using the photodiode labeled “PM” in Fig. S5 (ThorLabs S154C) and scaling it by the difference in attenuation between the optical path up to this photodiode (2 dB) and through the entire optical train (93 dB), 91 dB in total. These attenuation factors were measured at room temperature using the same photodiode, where the laser power is measured directly after the laser output, at the location of “PM” in Fig. S7, and directly after the aspheric lens. Since the maximum output of the Fabry-Perot laser is ~ 1.6 mW, the attenuated output power falls below the minimum resolvable optical power of the photodiode (10 pW). Therefore, to derive the total attenuation of the optical train, we remove the variable attenuator (ThorLabs VOA50-APC), separately characterize its attenuation (a maximum of 58.3 dB) and add this back to the attenuation of the optical train without the variable attenuator (35.7 dB). In principle, optical components that are cooled down will thermally contract and might exhibit additional insertion losses due to mechanisms like strain or misalignment. We measure this additional attenuation by measuring the laser reflectance signal of niobium at room temperature and at 20 mK. Assuming that the reflectance of niobium does not change when cooled down, there is no measurable difference between the reflectance signals at the two temperatures, implying that our optical train exhibits no additional attenuation when cooled down.”

Since we do not see any measurable variation in the reflectometry signal (and therefore observe no measurable change in the attenuation as the device is cooled), we consider the dominant source of error from the optical path to be fluctuations in the laser power. Below

we plot the normalized standard deviation in laser power for representative output powers of the laser. At low laser output powers we observe a 1.8% fluctuation in the output power. At higher laser powers (which is the standard operating range of the laser in this work), we observe <0.2% fluctuation in the output power. This would result in a $\pm 0.2\%$ error in the number of photons absorbed. With a $\eta = 87\%$, this would result in a value $\eta = 87\% \pm 0.17\%$.

- 2) In the introduction, the authors motivate the paper by saying “the energy gap provides a mechanism to distinguish photons from dark counts caused by fluctuations, but also limits the detection of lower-energy photons”. Superconducting detectors have established single photon detection far into the mid-infrared, out to 30 μm and beyond. This paper only demonstrates detection of 1550 nm photons. Can the authors comment on the low-energy limit of photons they believe this graphene-based device can detect?

We thank the reviewer for this comment. We applaud the capability of superconducting detectors that have established SPD far into the mid-infrared. Therefore, the longer wavelength photon would be the more promising application space for graphene single-photon bolometers. The proof-of-principle experiment to demonstrate the detection of a single photon by sensing the temperature rise of graphene electrons is the first step to achieve this long-term objective by demonstrating a fundamentally different detection mechanism that is broadly applicable to photons of different wavelengths.

The low-energy limit of graphene-based SPD remains at photons at about 10 GHz range, based on the theoretical and modeling from Phys. Rev. Applied 8, 024022 (2017), and the estimation from two back-to-back reports of graphene bolometer experiments: Nature 586,

42 (2020) and Nature 586, 47 (2020). The basic argument is that for a μm^2 size graphene flake at 10 mK, the total heat capacity of the graphene electrons is about a few k_B (Boltzmann constant). As a result, the temperature rise is on the order of a few 100 mK.

While the temperature rise is large compared with the base temperature, the challenge is that the thermal relaxation time to the electron-phonon channel can be very short, about 10 ns nanoseconds. Therefore, we need to upgrade the DC switching mechanism described in our current manuscript to the kinetic inductance readout scheme, first suggested in Appl. Phys. Lett. 92, 162507 (2008) and then implemented properly for graphene in Nano Lett. 23, 4136 (2023). This readout mechanism is analogous to the kinetic inductance detector – one should measure the reflection of a microwave tone off of the junction+resonator structure. In graphene-based detectors, however, when the photon is absorbed, the I_c of the device will decrease, which results in an increase in the Josephson inductance and a decrease in the resonance frequency. This shift will change the reflected probe tone power, which can be measured through a homodyne or heterodyne measurement, similar to the kinetic inductance detector.

Since the superconductor-graphene-superconductor is a proximity junction, its behavior depends on a lot of factors, including the choice of superconductor, cleanliness of graphene, the transparency of the contact, etc. In the past two years, we have been investigating different superconducting contacts to graphene to optimize the response of Josephson junction to a thermal change. To maximize the reflected signal from the kinetic inductance readout described above, one needs to consider the $dI_c/dT * I_c^{-1}$ of the device. We have performed a materials and design co-study measuring 6 different superconducting materials as leads and a wide range of Josephson junctions with different lengths, widths and across a broad range of densities and temperatures. We have submitted this optimization to peer-review publication [Jung et. al. arxiv 2503:86050 (2025)].

We are redesigning, fabricating, and starting to test new graphene single-photon bolometers that operate at 12 GHz, encouraged by the optimization and the proof-of-concept experiment under the current review. In summary, we believe the low-energy limit of graphene-based single-photon detectors is at about 10 GHz. We are overcoming the challenge of short integration time by kinetic inductance readout and optimization of the thermal response of Josephson junction.

- 3) The authors say on line 95 that they “register a click, reset the bias current” Can the authors further expand upon how this bias current is reset? Is it manually switched off upon detection? Is an inductive shunt or alternative low-resistance path to ground used to naturally divert the current?

We thank the author for this question. We believe the reset of the detector carries important information about the detector as Reviewer #1 also asked a similar question. In short, the reset the bias through an analog circuit, and not by a shunt circuit which is widely employed in the nanowire detectors. We provide the more detailed answer with the modified methods section to describe the analog circuit as follow:

“More specifically, the voltage from the device is referenced to a comparator and a DC voltage source. The DC voltage source is set well above the noise threshold of our digital to analog converter but below the normal state voltage. When a photon is absorbed, the junction switches, increasing the voltage triggering the comparator. The comparator sends a signal to a fast switch (response time much faster than the filter RC time) that sets $I_b < I_r$. The device then enters the superconducting state, where it has a lower voltage. Thus, it changes the signal of the comparator to switch the applied bias back to I_b .”

Please see the below figure from Evan Walsh’s Thesis (2020) (reference 45 in the main text) as a schematic.

- 4) The authors have a very in depth description the calculation of eta in the supplement, but very little description of how it is calculated in the main text. Adding further detail of how eta is calculated to the main text would be beneficial.

We agree with the reviewer that this description could be more detailed in the main. We have added a broad description of the calculation of eta into the methods section. It reads:

“Description of the Calculation of η In order to properly calculate η we need a thorough understanding of three factors 1) The measured switching rate, Γ_{meas} , 2) the amount of photon flux being sent by the laser and 3) the amount of photon flux absorbed by the graphene, which can be further split between a geometric component and the absorption of

the graphene within the van der Waals stack. The determination of Γ_{meas} was addressed in the previous section.

For (2), to calibrate the power incident on the graphene, we use a combination of live power monitoring and measured optical attenuation at room temperature. We monitor the live optical power using the photodiode labeled “PM” in Fig. S5 (ThorLabs S154C) and scaling it by the difference in attenuation between the optical path up to this photodiode (2 dB) and through the entire optical train (93 dB), 91 dB in total. These attenuation factors were measured at room temperature using the same photodiode, where the laser power is measured directly after the laser output, at the location of “PM” in Fig. S7, and directly after the aspheric lens. Since the maximum output of the Fabry-Perot laser is ~ 1.6 mW, the attenuated output power falls below the minimum resolvable optical power of the photodiode (10 pW). Therefore, to derive the total attenuation of the optical train, we remove the variable attenuator (ThorLabs VOA50-APC), separately characterize its attenuation (a maximum of 58.3 dB) and add this back to the attenuation of the optical train without the variable attenuator (35.7 dB). In principle, optical components that are cooled down will thermally contract and might exhibit additional insertion losses due to mechanisms like strain or misalignment. We measure this additional attenuation by measuring the laser reflectance signal of niobium at room temperature and at 20 mK. Assuming that the reflectance of niobium does not change when cooled down, there is no measurable difference between the reflectance signals at the two temperatures, implying that our optical train exhibits no additional attenuation when cooled down. Collectively, this process ensures a robust understanding of the photon flux reaching the mK stage of the fridge, N.

The photon flux absorbed by the graphene contains a geometric, and a thin film interference component. The geometric component describes how much overlap the graphene region has with the beam spot. In the supplementary information we show that we are able to estimate the size of the beam spot by using our reflectometry measurement to image known device features on the chip. Here, we assume the beam spot adheres to a Gaussian beam profile and we measure V_{refl} as the beam spot passes over a known feature. The image formed from V_{refl} is the convolution of the Gaussian beam spot and the feature. From here we are able to extract the beam waist and subsequently the proportion of the beam spot which overlaps with the graphene, α_{geo} (see Supplementary for more details).

The graphene absorption is found using the wave-transfer matrix method. This method accounts for the index of refraction of each layer of the van der Waals heterostructure and the subsequent reflection/transmission/absorption at each interface to extract the absorption into the graphene layer. In our devices, we find a graphene absorption $\alpha_{\text{gr}} \sim 0.6\%$ (see Supplementary for detailed calculation and relevant layer thicknesses).

In order to calculate the number of photons absorbed by the graphene we thus take $N\alpha_{\text{geo}}\alpha_{\text{gr}}$. From here, $\eta = \Gamma_{\text{meas}}/N\alpha_{\text{geo}}\alpha_{\text{gr}}$.

- 5) Line 171 “photon” should be “photons”

We thank the reviewer for this catch and the text has been fixed.

- 6) In Fig 2D, the V_{refl} peaks on the right side of the chart. It would be useful to denote in the chart itself that this is due to reflection off of the contact. Additionally, it would be useful to add a gray dashed line indication the device, to mirror the line in 2A, given that the vertical scale bars are different.

We thank the reviewer for this comment and we have updated the figure accordingly.

- 7) Line 232: “corresponding to $\eta \geq 0.8$ over I_b ranges $\sim 15\%$ of $\langle I_s \rangle$,” This sentence is unclear, do they mean within 15% of I_s ? If so, it may be more clear to say something along the lines of $>85\%$ of I_s .

We agree with the reviewer that the wording is unclear. We’ve changed the wording to: “corresponding to $\eta \geq 0.8$ for $I_b > 0.85 \langle I_s \rangle$.”

- 8) The authors don’t show any data of I_b for negative V_{Gate} . Is the photon detection behavior symmetric about the charge neutrality point?

This is a good question by the reviewer. From our Fig. S3C&D (reproduced below) it is clear that there is a robust supercurrent for negative gate voltages.

While commonly used contact metals (such as Al, NbN), often result in substantial doping of the graphene, the close workfunction to graphene of the MoRe contacts induce only a small amount of doping, enabling supercurrent to be mediated on either side of the Dirac peak (Borzenets et al PRL 117, 237002 (2016)). However, the critical currents are still reduced from this small amount of contact doping. In the case of both devices, this critical current was below 1 μ A, which is roughly the regime in which we were not able to resolve single photons for positive V_{gate} . As such the single photon behavior of the hole side of the Dirac peak is unexplored.

- 9) In the Table S2, the authors seem to use α_0 to denote the freestanding reflectance, but I can't find any place where this is explicitly stated.

The freestanding absorption of graphene, α_0 , is mentioned in words at the end of supplementary section B, however the symbol was not used. We have included the symbol in the revised version of the manuscript such that the sentence now reads:

"Using the measured thickness listed in Table X, we find that $\alpha_{\text{gr}} \sim 0.6\%$, lower than the free standing value, α_0 , of 2.3%"

- 10) In Figure 3B, how is I_r as a function of V_{gate} determined?

I_r is measured by biasing the device well-above the superconducting transition at I_s and reducing the bias current until it "retraps" into the superconducting state. We define I_r as the point at which the device is no longer the normal state resistance R_N . In the case of Figure 3B, this data is taken alongside the switching current measurements and so we define I_r as the average measured value of retrapping over thousands of sweeps. While there is some small spread in I_r , there is less variance as in I_s as the junction is heated in the normal state. This heating (in addition to junction damping effects) creates hysteresis in our IV curves (see below figure from Walsh et al Science 372 [2021]), which is the basis of the latching mechanism of our detector.

Upon looking at Figure 3B with this question in mind, we agree that the use of a dashed line makes the I_r seem like a calculated value. We have updated Figure 3B with 'dot dash' points to show that this was an experimentally obtained curve.

11) On Line 240, the authors say "For $V_{\text{gate}} < 0.5\text{V}$, the sharp suppression of I_s leads to a poor η ." The behavior of the device in this regime is unclear, and this sentence should be expanded upon qualitatively for clarity. Why doesn't it work near the charge neutrality point?

We thank the reviewer for this comment as the text was unclear. The highest η at 0.5V is not reduced, the bias range in which a high η can be achieved has been reduced. As such we have changed the text to the following:

"For $V_{\text{gate}} < 0.5\text{V}$, the sharp suppression of I_s leads to a reduction in I_{bias} for which $\eta > 0.8$. We attribute this observation to I_s approaching I_r , where the junction would have some probability of self-switching before latching could occur."

The DC switching mechanism that we use to register the event from a single photon arrival relies on the hysteresis of Josephson junction. When the effect of the hysteresis reduces near the Dirac point, that is the switching and retrapping currents become too close together, the mechanism can miss detecting the photon. This is because after the photon is absorbed and the junction switch to normal, the junction may retraps back to supercurrent state, without latching to the normal states for proper registration. This is evidenced by the reduction in the plateau width as the density approaches the Dirac peak. This problem could potentially be resolved by increasing the width of the Josephson junction, which would

increase the critical current of the device, however this would come at the cost of increasing the heat capacity due to the increased graphene area. Alternatively, shortening the length of the junction may increase the critical current, however this would be somewhat marginal.

Further, we thank the reviewer for this question as it pointed us to a minor error in Figure 3A. The left-most curve was labeled as “0.25 V”, when in fact it should read “0 V”. The curves for 0 V and 0.25 V are plotted below. We can in fact see the single photon detection at 0.25 V, albeit in a limited bias range such as the range of I_b from 1.2 - 1.3 μA in the blue data at 0.25 V in the figure below.

- 12) The thicknesses of the graphene and hBN layers determine the reflectances calculated in the wave-transfer matrix. Table S2 lists the thicknesses of the materials, but it is unclear how these thicknesses are measured. Specifically the bottom graphite will have a large impact on the reflectance of the device, and it is unclear how confident they authors are they have 1.33 and 2 nm thickness for d4.

Figure S1. AFM topography images for two devices. (a) Optical microscope images of the stacks for device A (left) and device B (right). The magenta line marks the graphene, while the dashed line marks the graphite. AFM topography was measured in the regions outlined by the black boxes. Scale bar: 10 μm . (b) AFM topography images of the areas indicated by the black boxes in (a), for device A (left) and device B (right). (c) Line profiles extracted from (b) along the dashed lines: blue (device A, left) and red (device B, right).

We thank the reviewer for this comment and have added the above figure and the following discussion to the supplementary information:

“We determined the thicknesses of the top and bottom hBN layers from AFM topography

measurements on the stack, as shown in Figure S1. In contrast, the bottom graphite thickness cannot be precisely extracted from AFM due to the limited resolution and the presence of surface impurities on the stack. To address this, we quantified the optical contrast of the graphene region relative to the bare substrate by analyzing the average intensity of the green channel in optical microscope images. This approach follows the method established by Ni *et al.* (*Nano Letters* **2007**, *7*, 2758–2763) and Wang *et al.* (*Nanotechnology* **2012**, *23*, 495713), who demonstrated a clear, quantitative correlation between the number of graphene layers on a SiO₂ (≈ 285 nm)/Si substrate and the green-channel contrast. For example, Wang *et al.* reported that the green contrast increases systematically with thickness: 7.7 %, 14.9 %, 21.6 %, and 27.8 % for 1,2,3, and 4 layer graphene, respectively. Using this calibration as shown in Figure S2, we assigned the bottom graphite thickness in device A to 6 layers (≈ 2 nm, 21.4 %) and in device B to 4 layers (≈ 1.33 nm, 17.1 %).

Figure S2. Green-channel optical contrast as a function of the number of graphene layers on a SiO₂ (≈ 285 nm)/Si substrate.

Reviewer #3 (Remarks to the Author):

Review of the manuscript by B. Huang et al., “Thermal Detection of Single Photons Using Dirac Fermions”

The manuscript reports on the development of a single infrared photon counter based on a graphene absorber integrated with a graphene-based Josephson junction. The authors’ central idea is to spatially separate the photon absorption site—where the absorption event raises the electron temperature in graphene—from the detection site, where the resulting temperature rise is sensed via the switching of the Josephson junction into the resistive state.

The small heat capacity of graphene at low carrier density, close to the Dirac point, enables a pronounced temperature increase upon photon absorption, which in turn enhances device sensitivity. The spatial separation between the absorber and the Josephson junction also mitigates detrimental effects of high-energy photons on the superconducting properties of the junction leads.

The experimental results convincingly support the proposed detection mechanism. The authors present clear evidence that the device response peaks at the Dirac point and systematically evaluate the response while scanning the photon beam spot across the graphene absorber. These measurements are complemented by a solid theoretical analysis of heat propagation along the graphene strip, yielding a coherent and consistent picture of the device operation.

However, I see at least two major points for improvement of the manuscript in its current version, these are: a) Lack of performance benchmarking, and b) 2. Clarification of application scope – these points are elaborated below:

1) Lack of performance benchmarking

The manuscript does not compare the demonstrated infrared photon counter with the state-of-the-art performance of similar devices. Given that near-infrared single-photon detectors are a key enabling technology for quantum information science, the field is highly competitive. Several relevant works—absent from the current reference list—should be cited and discussed, including:

- o R. E. Warburton et al., *Optics Letters* 32, 2266 (2007)
- o W. Guo et al., *Applied Physics Letters* 110, 212601 (2017)
- o I. Craiciu et al., *Optica* 10, 183 (2023)

A quantitative comparison of key performance metrics (efficiency, timing jitter, dark count rate) with these and other leading technologies would allow readers to assess the novelty and competitiveness of the reported device.

We thank the reviewer for this comment and we agree that positioning our device with other 1550 nm detection schemes is critical for assessing the performance of our device. As such, we have added a table in the supplementary information comparing the suggested key metrics to others in the field. This highlights that our detector is comparable to state-of-the-art SNSPDs with respect to quantum efficiency for a given dark count rate (as highlighted by our low NEP value). While our count rate is not competitive, this is verifiably due to the 30 kHz external RC filters used in the measurement (as explained elsewhere in the review and in the new section of the supplementary information titled “Measurement Bandwidth and Photon Flux Considerations”). We note that the count rate as implemented in this proof-of-principle experiment is far-below the possible limit by the detection

mechanism, bounded by the convolution of thermal relaxation and readout resonator bandwidth.

TABLE S6. Comparison of Detector Platforms. Efficiencies marked with a *, denote system efficiency.

Work	Platform	Efficiency	DCR	Timing Jitter	Count Rate
This Work	GJJSPD	87%	1/s	2.7 ps [46]	1.5 kc/s
Ref. [68]	SNSPD	78%	158/s	50 ps	1.5 Gc/s
Ref. [69]	SNSPD	1%	100/s	68 ps	100 Mc/s
Ref. [70]	SNSPD	28%	$6 \times 10^{-6}/s$	—	—
Ref. [71]	SNSPD	93%*	1/s	150 ps	25 Mc/s
Ref. [72]	mKID	0.38%	—	—	—
Ref. [73]	SPAD	50%*	20000/s	70 ps	1 Mc/s

2) Clarification of application scope

The manuscript briefly mentions potential applications to axion searches. However, the photon energy range relevant to such experiments (10^{-6} – 10^{-3} eV, corresponding to frequencies below ~ 0.25 THz) is far below the ~ 200 THz photon energies studied here. This significant mismatch should be explicitly acknowledged. (Here I refer to a comprehensive review by Bradley, et al., ‘Microwave cavity searches for dark-matter axions’, Rev. Mod. Phys 75, 777 (2003))

In this context, the authors should also reference recent work directly targeting the relevant frequency range, such as:

- o L. Pankratov et al., Nature Communications 16, 3457 (2025), which reports single-mode thermal microwave photon detection at 30 GHz using a similar superconducting sensor and currently sets the state-of-the-art in that domain.

We agree with the reviewer that the application scope is an important subject in the development of the single-photon detector. As such we have softened the language in the abstract from: ‘opening pathways to study space science in far-infrared regime, to search for dark matter axions,’ to ‘to potential applications in dark matter searches’.

Regarding the recent work by Pankratov et al. (Nature Communications 16, 3457 [2025]), we appreciate the reviewer bringing this excellent result to our attention. However, our manuscript focuses specifically on the fundamental physics of phonon-free single-photon detection using Dirac fermions. Given the substantial recent progress in microwave single-photon detection—including approaches based on qubits [PRX 6, 031036 (2016)], λ -systems [Nature Comm. 7, 12303 (2016)], and dissipative engineering [Phys. Rev. X 10, 021038 (2020)] at GHz frequencies, we believe maintaining this focused scope serves the reader best. We are currently preparing a separate manuscript within which we have a brief

survey of various single-photon detectors for dark matter searches, where the Pankratov et al. work will certainly be featured as an important reference.

Summarizing, the manuscript presents a technically sound and carefully executed study, supported by both experimental and theoretical analysis. The work is of interest to the community working on graphene-based superconducting devices and photon detection. However, in its current form, it does not convincingly demonstrate performance beyond the state-of-the-art, nor does it provide a sufficiently detailed comparison with competing technologies.

Given these limitations, I do not recommend publication in Nature Communications at this stage. A strengthened manuscript should include a thorough benchmarking against existing devices and a clearer discussion of realistic application domains.